# Financing of Immunization Programs by Local Government Units in Poland as an Element of Health Policy

**DOI:** 10.3390/vaccines10010028

**Published:** 2021-12-27

**Authors:** Anna Augustynowicz, Mariola Borowska, Katarzyna Lewtak, Jacek Borowicz, Michał Waszkiewicz, Beata Karakiewicz, Janusz Opolski, Tomasz Banaś, Aleksandra Czerw

**Affiliations:** 1Department of Health Economics and Medical Law, Medical University of Warsaw, 02-091 Warsaw, Poland; anna.augustynowicz@wum.edu.pl (A.A.); aleksandra.czerw@wum.edu.pl (A.C.); 2Centre of Postgraduate Medical Education, School of Public Health, 01-81 Warsaw, Poland; michal.waszkiewicz@cmkp.edu.pl; 3National Institute of Public Health NIH—National Research Institute, 00-791 Warsaw, Poland; klewtak@pzh.gov.pl; 4Department of Prevention of Environmental Hazards and Allergology, Medical University of Warsaw, 02-091 Warsaw, Poland; jacek.borowicz@wum.edu.pl; 5Department of Social Medicine and Public Health, Pomeranian Medical University, 70-204 Szczecin, Poland; beata.karakiewicz@pum.edu.pl; 6Faculty of Engineering and Management, University of Ecology and Management, 00-792 Warsaw, Poland; januszopolski@wseiz.edu.pl; 7Department of Gynecology and Oncology, Faculty of Medicine, Jagiellonian University Medical College, 31-007 Krakow, Poland; tbanas@mp.pl; 8Department of Economic and System Analyses, National Institute of Public Health NIH—National Research Institute, 00-791 Warsaw, Poland

**Keywords:** immunization, health policy programmes, primary prevention

## Abstract

Introduction: The scope and schedule of immunization in Poland is regulated by the Immunization Programme prepared and announced by the State Sanitary Inspector. There are two kinds of vaccines: compulsory vaccines, financed by the state budget at the disposal of the Minister of Health, and vaccines recommended by the central health authorities but financed by local governments within health policy programmes. Compulsory vaccines cover people up to 19 years of age and individuals at higher risk of infections. The public health programmes organized and financed by local governments play an important role in infectious disease control in the country. Objective: The objective of this study is to analyse health policy programmes including immunization programmes, which were developed, implemented and financed by local government units of all levels in Poland between 2016 and 2019. Material and Methods: This analysis covers data compiled by voivodes and submitted to the Minister of Health as annual information on public health tasks carried out by local government units. From the aggregate information, data on all health policy programmes conducted by individual local government units between 2016 and 2019, including immunization, were extracted and analysed. The data were obtained pursuant to the provisions of the act on access to public information. Results: In the analysed period, local government units implemented a total of 1737 health policy programmes that financed the purchase of vaccines, qualification tests for immunization and carrying out immunization by authorized medical entities. Among the vast majority of programmes, promotional activities were also implemented. Conclusions: In Poland, local governments are deeply engaged in the immunization of their citizens by organizing and financing specific health care programmes. These programmes are an essential addition to the state financial resources in infectious disease control. This engagement expresses local government maturity regarding the health needs of the population and public health measures. Communes are the most engaged units among all levels of local governments. It is probably due to close mutual communication between the people and local governments. The growing awareness of the important role of HPV immunization in the prevention of cervical cancer among local government units is reflected in the increase in the number of girls vaccinated against HPV and the increase in financial resources allocated for primary HPV prevention. The decrease in the number of people vaccinated against pneumococci may result from including pneumococcal vaccines in the compulsory immunization schedule.

## 1. Introduction

Immunization is carried out in Poland in accordance with the Immunization Programme announced by the Chief Sanitary Inspector [1]. The programme specifies compulsory vaccines and recommended vaccines. Compulsory vaccines are financed from the state budget funds at the disposal of the Minister of Health. Immunization is carried out according to the immunization schedule in people up to 19 years of age and groups at risk of infections, including students of medical universities, health workers, and employees of veterinary services. The information collected by employees of sanitary and epidemiological stations on the implementation of compulsory immunization in Poland shows that, in recent years, 95% of children and adolescents were subject to compulsory immunization. The implementation of the immunization programme resulted in a significant reduction in or elimination of diseases such as diphtheria, tetanus, polio, measles, rubella, hepatitis B, or infections caused by *Hemophilus influenzae* [2].

The list of recommended vaccines is also specified in the Immunization Program. Recommended vaccines and vaccines for people who are not subject to compulsory immunization are financed from the household budget. They can also be financed or co-financed by local government units. These programmes are devoted specifically to immunization, or immunization issues are part of broader health programme. These programmes are important in combating infectious diseases in Poland. They are also a vital element in the development of self-governance and citizens’ responsibility for their health.

## 2. Objective

The objective of this study is to analyse health policy programmes, including immunization, which were developed, implemented and financed by local government units of all levels in Poland between 2016 and 2019.

## 3. Methods and Material

The study was conducted based on the analysis of existing data. Data from annual information prepared by voivodes (voivodes are representatives of the Government of the Republic of Poland in individual voivodships, and one of the statutory tasks assigned to voivodes is collecting, analysing and submitting data on public health tasks performed by local government units to the Minister of Health) and submitted to the Minister of Health on public health tasks performed by local government units were used (at the national level, activities in the field of public health are primarily carried out by the Minister of Health as a representative of the government administration, and at the regional and local levels, these tasks are carried out by local government units) [3]. Data on all health policy programmes conducted by individual levels of local government units, including immunization, were extracted from the collective information [4]. The analysis covered programmes completed by local government units of all ranks, i.e., by the largest administrative units (i.e., self-governments of voivodeships), by second-degree administrative units that rank below voivodeships (i.e., counties) and by primary administrative units (i.e., municipalities). The class of counties also includes county towns. The data were obtained pursuant to the provisions of the act on access to public information.

Information was presented on the number of local government units implementing these programmes, the number of vaccinated people and the total cost of the programmes between 2016 and 2019. Data were also provided on the main types of vaccines carried out by local government units as part of health policy programmes, i.e., against influenza, HPV (*Human Papilloma virus*), pneumococci (*Streptococcus pneumoniae*) and meningococci (*Neisseria meningitidis*). In this area, data were presented on the total cost of programmes and the share of expenditure on immunization programmes in total health policy programmes, the number of people who were vaccinated and the percentage of vaccinated persons in the target population. Information on the unit cost of HPV, influenza, pneumococcal and meningococcal vaccines is also presented.

## 4. Results

Between 2016 and 2019, local government units implemented a total of 1737 health policy programmes under which immunization was carried out. The programmes financed the purchase of vaccines, qualification tests for immunization and immunization carried out by authorized medical entities. The vast majority of programmes also carried out activities promoting the idea of immunization by financing advertising spots, educational and information materials, etc.

Data on the number and percentage share of local government units implementing health policy programmes, the number of all public health tasks implemented by local government units, the number of vaccinated people and the total cost (the total cost of programs in USD in individual years is presented on the basis of the average USD exchange rate in individual years, which in 2016 was PLN 3.9211, in 2017 was PLN 3.7957, in 2018 was PLN 3.6117 for USD 1.00, and in 2019 was PLN 3.8399 for USD 1.00) of programmes implemented between 2016 and 2019 are presented in Table 1.

Between 2016 and 2019, local government units implemented a total of 1737 programmes. A total of USD 30,270,688 was spent on their implementation, and 1,080,012 people were vaccinated under health policy programmes.

In the analysed period, eight voivodship self-governments implemented 32 programmes under which immunization was carried out. Voivodship self-governments allocated USD 4,831,168 for the implementation of the programmes. Slightly over 47% of this amount, i.e., USD 2,285,893, was spent in 2019, and 35,889 people were vaccinated under the programmes implemented between 2016 and 2019 by voivodeship self-governments. In each subsequent year, the amount of funds allocated to preventive vaccines increased and so did the number of people covered by vaccines.

Poviats implemented 209 programmes under which immunization was carried out. Poviats allocated USD 5,198,097 to their implementation. The vast majority of this amount, i.e., 67.2% (USD 3,491,943), was spent in 2016. The lowest amount was allocated to the implementation of immunization programmes in 2017 (USD 229,810). During these programmes, 147,007 people were vaccinated, with the largest number in 2016 (101,400 people) and the smallest in 2017 (8392 people).

Communes implemented 1103 programmes. The total cost of the programmes was USD 20,170,431. Most funds, i.e., 48.5% of the total amount (USD 9,786,894), were allocated to immunization programmes in 2019. During the period under review, 881,027 people were vaccinated. In each subsequent year, the amount of funds allocated to immunization increased. Between 2016 and 2018, the number of programmes implemented by communes increased and so did the number of people vaccinated under the programmes.

Out of 1737 programmes, 38.7% (673 programmes) were HPV immunization programmes. Influenza immunization programmes accounted for 36.2% (628 programmes) of all programmes implemented. There were also 283 (16.3%) pneumococcal immunization programmes and 132 (7.6%) meningococcal immunization programmes. Between 2016 and 2018, the number of programmes in each of the groups mentioned increased. During this period 138, 158 and 235 HPV immunization programmes; 108, 112 and 152 influenza immunization programmes; 43, 92 and 115 pneumococcal immunization programmes; and 27, 31 and 38 meningococcal immunization programmes were implemented. In 2019, the number of programmes in each group decreased. In 2019, 142 HPV immunization programmes, 193 influenza immunization programmes, 33 pneumococcal immunization programmes and 36 meningococcal immunization programmes were implemented. Between 2016 and 2019, twenty-one programmes were also carried out, including hepatitis B, tick-borne encephalitis, varicella and rotavirus immunization programmes.

Information was provided on the number and percentage of people vaccinated against HPV, influenza, pneumococci and meningococci in Poland and the average percentage of people vaccinated in the European Union between 2016 and 2019 (Table 2).

The general level of immunization of the target populations in Poland between 2016 and 2019, in the case of each of the analysed types of vaccines, differed—to a detriment—from the average level of immunization in the European Union.

In Poland, in each year of the period discussed, approximately 2.4% of the target population (i.e., girls aged 10–14) was covered by vaccines against HPV; in the European Union, this percentage was ten times higher and, in each year of the analysed period, on average, was around 24%.

The average influenza immunization of the target population in the EU between 2016 and 2019 was around 36.4%; in Poland, the percentage was nearly fifteen times lower: only around 2.5% in the indicated time frame.

The level of immunization against pneumococci in Poland in individual years between 2016 and 2019 did not exceed 1% and decreased each year. In the European Union, the percentage of people vaccinated against pneumococci in the period discussed increased each year from 56% in 2016 to 80% in 2019.

The lack of information on the level of immunization against meningococci in the target population does not allow for the determination of differences between the immunization level in Poland and the average immunization level in the European Union.

The table three presents information on the expenses incurred by local government units on programmes including immunization against HPV, influenza, pneumococci and meningococci and their percentage in total health policy programmes implemented between 2016 and 2019. (Table 3)

Between 2016 and 2019, local government units allocated the most financial resources to influenza immunization programmes. USD 9,304,138 was spent on influenza immunization programmes. Voivodship self-governments spent USD 60,563, poviats spent USD 2,496,935 and communes spent USD 6,746,640. Communes allocated about 72.5% of the amount allocated for this purpose by local government units at all levels for immunization.

USD 8,417,766 was spent on HPV immunization programmes between 2016 and 2019. Voivodship self-governments spent USD 1,790,941, poviats spent USD 1,075,517 and communes spent USD 5,551,308. Communes allocated about 66% of the amount allocated for this purpose by local government units at all levels for these programmes.

USD 4,803,275 was spent on pneumococcal immunization programmes. Voivodship self-governments spent USD 1,472,690, poviats spent USD 1,274,264 and communes spent USD 2,056,321. Communes spent 42.8% of the funds allocated for this purpose by local government units at all levels.

USD 1,258,547 was spent on meningococcal immunization programmes. Voivodship local governments did not implement meningococcal immunization programmes in the period discussed. Poviats allocated USD 124,152 for this purpose, while communes allocated USD 1,134,394. Communes spent over 90% of the funds allocated to meningococcal immunization programmes by local government units at all levels.

Data on the number of people vaccinated against HPV, influenza, pneumococci and meningococci within the programmes and the percentage of people vaccinated within the programmes in the target population between 2016 and 2019 are presented in Table 4.

Between 2016 and 2019, under immunization programmes, the largest number of people were vaccinated against influenza (725,093 people). In the discussed period, 153,020 people were vaccinated against pneumococci, 118,908 girls were vaccinated against HPV and 28,780 people were vaccinated against meningococci. In all types of vaccine, the largest number of people was vaccinated under programmes implemented by communes and the smallest number was vaccinated under programmes implemented by voivodship local governments.

The highest percentage of the target population (i.e., girls 10–14 years old) was vaccinated against HPV. The highest percentage of the target population was vaccinated against HPV through programmes implemented by communes. For programmes that include other vaccines, this percentage does not exceed 0.65% of the target population.

Table 5 presents information on the unit cost (in USD) of HPV, influenza, pneumococcal and meningococcal vaccines between 2016 and 2019.

The unit cost of HPV vaccines increased in each subsequent year, ranging from USD 36.13 in 2016 to USD 110.53 in 2019. The unit cost of pneumococcal vaccines was the highest in 2016, being USD 105.88, and the lowest in 2017, being USD 8.67.

## 5. Discussion

The presented results indicate that, between 2016 and 2019, some local government units were involved in the implementation of health policy programmes including immunization. In 2019, such activities were carried out by half of the voivodship self-governments, every 11th poviat and every 9th commune. Taking into account the fact that the development, implementation and financing of health policy programmes are optional tasks of local government units, undertaking actions in this area should be assessed positively. This is an expression of the maturity of local government units in the approach to preventing infections and infectious diseases.

Voivodship local governments and communes in each subsequent year increased the number of immunization programmes, the amount of financial resources allocated to their implementation and the number of people covered by vaccines, or maintained them at a similar level. It was all the more important as the situation in Poland in comparison with the EU in terms of vaccines against HPV, influenza and pneumococci is much worse, which is shown in the data presented in the manuscript. No such regularity was observed in the case of programmes implemented by poviats. The diversification of poviats’ engagement in the implementation of programmes in individual years may be caused by many different reasons, e.g., the economic situation of individual local government units. It is also possible that poviats allocated, in certain years, more financial resources to other activities in the field of health protection, e.g., financing the provision of therapeutic health services or financing investments in health care. Identifying the reasons for this diversification, which is extremely important in the context of the premises for health policy, would require additional, targeted research.

Communes were most active in the implementation of health policy programmes involving immunization. Among local government units of all levels, the largest number of people were vaccinated by communes in all vaccine types discussed, allocating the highest financial resources for this purpose. It cannot be ruled out that this is the result of a closer bond between the inhabitants of a commune and local authorities; specific mutual communication; and thus, greater activity, also in the area of health.

The results obtained indicate that local government units were most willing to engage in the implementation of influenza immunization programmes. In Poland, the influenza vaccine is a recommended vaccine. During the study period, people under 65 years of age could purchase the vaccine at its full price. In turn, from July 1, 2018, people over 65 could get a 50% refund for their payment [5].

Taking into account the fact that, in Poland, the percentage of the population vaccinated against influenza is at a very low level, undertaking initiatives in this regard by local government units should be assessed positively. At the same time, local government units should be recommended to engage in promoting these vaccines.

Local government units were also involved in the financing of HPV vaccines. The increasing number of people vaccinated against HPV each year and the increasing total amount of financial resources allocated to the implementation of the programmes should be positively assessed. The increasing total number of vaccinated people and the increasing total amount of financial resources involved in the implementation of the programmes in each subsequent year should be assessed positively. It cannot be ruled out that this is due to the increased awareness of the importance of HPV immunization in the cervical cancer control strategy. At this point, it is worth noting that, in Poland, organized activities aimed at combating malignant cervical cancer are carried out within the framework of the National Programme for Combating Cancer implemented throughout the country [6,7]. Moreover, activities in the field of primary and secondary prevention of cervical cancer are carried out by local government units within the framework of health policy programmes. HPV immunization is carried out as part of primary prevention. In Poland, these vaccines are not compulsory and are financed from household budgets or as part of health policy programmes.

Nevertheless, in the context of the epidemiological situation, these activities should be considered insufficient. Malignant cervical cancer is one of the most frequently registered cancers in women in Poland [8]. Cervical cancer mortality in Poland is twice as high as in the EU27 states [9]. In Poland, one of the lowest 5-year survival rates in Europe is recorded. This percentage for patients diagnosed in 2000–2002 was 54.1% [10] with the European average of 62.1% [11]. One of the directions of activities should be the introduction of a universal HPV immunization programme financed from public funds. At present, HPV vaccines in Poland are recommended by the Chief Sanitary Inspectorate, but they are not financed from public funds. This would be in line with the guidelines of the World Health Organization (WHO). According to the WHO position, HPV immunization is recommended as one of the elements of the national strategy of reducing the incidence, morbidity and mortality associated with cervical cancer [12,13]. At this point, it is important to point out the importance of activities aimed at promoting the idea of HPV immunization activities that should be undertaken not only by local government units but also by government administration and non-governmental organizations.

Local government units financed pneumococcal and meningococcal immunization programmes. Until 31 December 2016, pneumococcal vaccines were available in Poland only as fully paid for the recipient or through health policy programmes implemented by local government units. As of 1 January 2017, the pneumococcal vaccine was entered into the Immunization Programme as a compulsory vaccine in the general population of children and covers children born after 31 December 2016. The presented data indicate a clear decrease in the number of people vaccinated in 2018 and 2019 in programmes implemented by communes. In the case of programmes run by voivodeship self-governments, the decrease occurred in as early as 2017. It is probably a consequence of the abovementioned change. It is also worth noting that pneumococcal vaccines for adults, including seniors over 65 years of age and people born before 31 December 2016, were at that time and still are among the recommended paid vaccines. Therefore, the efforts of voivodeship and commune self-governments should focus mainly on the immunization of as many of these people as possible against pneumococci. No such regularity was observed in the case of programmes implemented by poviats. In this case, the number of vaccinated people and expenditure on programmes grew until 2019. In 2019, a slight decrease was observed in this regard. It is possible that, since 2017, poviats were much more effective than the remaining levels of local government units in reaching people for whom immunization was not financed from public funds. Hence, the number of vaccinated people increased.

Activities of local government units in the field of financing immunization as well as promoting the idea of immunization should complement the activities undertaken by the state in this area and in the context of people becoming acquainted with the vaccines in question, and consequently, the percentage of vaccinated people should gradually increase.

## 6. Conclusions

In Poland, local governments are deeply engaged in the immunization of their citizens by organizing and financing specific health care programmes. These programmes are an essential addition to the state financial resources in infectious disease control.This engagement expresses local governments maturity regarding the health needs of the population and public health measures.Communes are the most engaged units among all levels of local governments. It is probably due to close mutual communication between people and local governments.The growing awareness of the important role of HPV immunization in the prevention of cervical cancer among local government units is reflected in the increase in the number of girls vaccinated against HPV and the increase in financial resources allocated for primary HPV prevention.The decrease in the number of people vaccinated against pneumococci may result from including pneumococcal vaccines in the compulsory immunization schedule.

## Figures and Tables

**Table 1 vaccines-10-00028-t001:** Immunization carried out by local government units between 2016 and 2019.

Year		Voivodeship Self-Government	Poviat	Commune
2016	number of LGUs implementing the program percentage of all LGUs at a given level	2 12.5%	44 11.6%	173 7.0%
	number of immunization programmes	4	91	243
	number of vaccinated people	3386	101,400	64,667
	total cost of programmes (in USD)	256,693	3,491,943	1,419,373
2017	number of LGUs implementing the program percentage of all LGUs (at a given level)	4 25.0%	24 6.3%	264 10.7%
	number of immunization programmes	5	27	327
	number of vaccinated people	7423	8392	280,184
	total cost of programmes (in USD)	817,936	229,810	2,746,479
2018	number of LGUs implementing the program percentage of all LGUs (at a given level)	8 50.0%	40 10.5%	384 15.5%
	number of immunization programmes	12	49	555
	number of vaccinated people	12,443	17,951	294,009
	total cost of programmes (in USD)	1,503,866	775,235	6,345,290
2019	number of LGUs implementing the program percentage of all LGUs (at a given level)	8 50.0%	34 8.9%	282 11.4%
	number of immunization programmes	11	42	371
	number of vaccinated people	12,637	19,264	242,167
	total cost of programmes (in USD)	2,285,893	611,276	9,786,894

**Table 2 vaccines-10-00028-t002:** People vaccinated in Poland and the EU between 2016 and 2019.

	2016		2017		2018		2019		EU *
	*n*	%	*n*	%	*n*	%	*n*	%	%
HPV	22,710	2.6%	19,961	2.2%	22,341	2.4%	25,079	2.5%	23% (2016) 24% (2017) 25% (2018) 25% (2019)
influenza	857,029	2.2%	945,869	2.5%	1,009,285	2.6%	1,013,706	2.6%	34.6% (2016) 35.6% (2017) 37% (2018) 38.6% (2019)
pneumococci	240,472	0.63%	185,966	0.48%	188,291	0.49%	30,083	0.08%	56% (2016) 73% (2017) 78% (2018) 80% (2019)
meningococci	54,863	0.14%	55,847	0.15%	98,223	0.26%	98,952	0.26%	**

* ECDC data. ** no information is available on the percentage of people vaccinated against meningococci in the EU in individual years.

**Table 3 vaccines-10-00028-t003:** Expenses on immunization programmes implemented between 2016 and 2019.

		HPV		Influenza		Pneumococci		Meningococci	
		Expenses (in USD)	%	Expenses (in USD)	%	Expenses (in USD)	%	Expenses (in USD)	%
2016	voivodeship	0	0.0%	0	0.0%	256,693	100.0%	0	0.0%
	poviat	266,490	7.6%	2,213,446	63.4%	1,012,894	29.0%	16,569	0.5%
	commune	548,326	38.5%	415,305	29.3%	309,966	21.8%	151,922	10.7%
2017	voivodeship	39,185	12.1%	0	0.0%	284,770	34.8%	0	0.0%
	poviat	139,293	58.3%	28,290	12.3%	45,375	19.7%	26,041	11.3%
	commune	888,566	30.8%	1,288,853	46.9%	633,673	23.1%	72,398	2.6%
2018	voivodeship	383,978	39.0%	26,951	1.8%	573,790	38.2%	0	0.0%
	poviat	322,190	50.8%	135,330	17.5%	139,402	18.1%	37,105	4.8%
	commune	1,611,080	28.1%	2,683,002	42.3%	967,246	15.2%	471,826	7.4%
2019	voivodeship	1,367,777	77.8%	33,611	1.5%	357,437	15.6%	0	0.0%
	poviat	347,544	59.1%	119,869	19.6%	76,593	12.5%	44,437	7.3%
	commune	2,503,336	46.0%	2,359,479	11.2%	145,435	0.7%	438,248	2.1%

**Table 4 vaccines-10-00028-t004:** People vaccinated under immunization programmes between 2016 and 2019.

		HPV		Influenza		Pneumococci		Meningococci	
		Number	% of the Population	Number	% of the Population	Number	% of the Population	Number	% of the Population
2016	voivodeship	0	0.0%	0	0.0%	3386	0.009%	0	0.0%
	poviat	7623	0.863%	83,644	0.218%	8940	0.023%	617	0.002%
	commune	14,927	1.690%	40,509	0.105%	2592	0.007%	4466	0.012%
2017	voivodeship	814	0.090%	0	0.0%	3609	0.009%	0	0.0%
	poviat	3271	0.361%	3230	0.008%	1423	0.004%	468	0.001%
	commune	20,829	2.296%	148,950	0.388%	106,138	0.276%	2511	0.007%
2018	voivodeship	2217	0.234%	1512	0.004%	5714	0.015%	0	0.000%
	poviat	4780	0.505%	8170	0.021%	1678	0.004%	1357	0.004%
	commune	26,277	2.776%	232,456	0.605%	12,423	0.032%	8610	0.022%
2019	voivodeship	3513	0.357%	1450	0.004%	4303	0.011%	0	0.0%
	poviat	7495	0.761%	8605	0.022%	1068	0.003%	1502	0.004%
	commune	27,160	2.758%	196,567	0.512%	1746	0.005%	9249	0.024%

**Table 5 vaccines-10-00028-t005:** Unit cost (in USD) of vaccines between 2016 and 2019.

	2016	2017	2018	2019
HPV	36.13	42.83	69.64	110.53
influenza	21.17	8.66	11.75	12.16
pneumococci	105.88	8.67	84.81	81.42
meningococci	33.15	33.04	51.06	44.90

## Data Availability

Data in the tables was compiled on the basis of information from the Ministry of Health. In table two we take information about EU from: https://www.ecdc.europa.eu/sites/default/files/documents/influenza-vaccination-2007%E2%80%932008-to-2014%E2%80%932015.pdf; https://immunizationdata.who.int/pages/coverage/hpv.html?CODE=eur&ANTIGEN=&YEAR=, https://immunizationdata.who.int/pages/coverage/pcv.html?CODE=eur&ANTIGEN=&YEAR=, https://www.ecdc.europa.eu/sites/default/files/documents/influenza-vaccination-2007%E2%80%932008-to-2014%E2%80%932015.pdf, and about Poland from: http://wwwold.pzh.gov.pl/oldpage/epimeld/2019/Sz_2019.pdf, http://wwwold.pzh.gov.pl/oldpage/epimeld/2018/Sz_2018.pdf, http://wwwold.pzh.gov.pl/oldpage/epimeld/2017/Sz_2017.pdf, http://wwwold.pzh.gov.pl/oldpage/epimeld/2016/Sz_2016.pdf, The population information used to calculate the proportion of vaccinated persons in the population is from the website of the Central Statistical Office: https://stat.gov.pl/obszary-tematyczne/ludnosc/ (all accessed on 12 December 2021).

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
