# Peer review of "Financing of Immunization Programs by Local Government Units in Poland as an Element of Health Policy"

_vaccines, 2021, doi:10.3390/vaccines10010028_

Round 1
Reviewer 1 Report
- The statistical table is not standard.There are too many horizontal lines in the statistical table; some statistical tables have cross-page design problems; percentage units appear in the table body.
- It is suggested to increase statistical charts, so that different vaccines can be compared between different years.
Author Response
Modifications were made to the tables - standard tables were used.
Reviewer 2 Report
In the abstract, it would be good to inform the readers what "voivodes" are. Also good to italicise Haemophilus influenzae at the end of the first paragraph of the Introduction as well as Streptococcus pneumoniae and Neisseria meningitidis further on. Would also be good to define "poviats" at Line 78.
Material is introduced in the Discussion which has not been presented in the Results: "What's important, maintaining a high rate of vaccinated people not only in the entire population, but also in local communities, with particular emphasis on disadvantaged groups with low socio-economic status, is essential to reducing social inequalities in terms of health". If this is a stated aim of the vaccination programme, it would be worth including that in the Results or at least the Introduction.
The four conclusions are central to the analysis. It would be good to include those in the abstract.
I am not sure that the article is of interest outside Poland.
- The main question addressed is: "To analyse health policy programs including immunization programs, which were developed, implemented and financed by local government units of all levels in Poland between 2016 and 2019".
- The topic is certainly original and relevant in Poland. Polish readers, especially those with an understanding of health policy programs in Poland, will benefit a great deal. The use of the phrase: "health policy programs including immunization programs" implies that the study examines ALL health policy programs in Poland, including those that cover immunization. It needs to be changed to simply: "To analyse health policy immunization programs". Unfortunately, for non-Polish readers, all the data about number vaccinated in vovoideships, poviats and communes is rather hard to comprehend - one doesn't know what these actually are. They are described in lines 73 to 79 but maybe it would help to know how many of each there are at each level, in the country. We aren't told exactly who decrees all the programs - do the decree come down from central government or are they decreed at the local level?
- The material certainly adds to existing knowledge, but one needs to do internet searches to establish exactly what the immunisation schedule is in Poland and when vaccines were introduced, and for whom.
- There needs to be no change in the methodology. Just a description of the Polish immunisation schedule, when vaccines were introduced and for whom. Maybe an appendix.
- The conclusions are appropriate.
- The references are appropriate.
- No comments on the tables.
Author Response
1 Note included
2 Note included
3 Note included
4 Note included. Removed this section from the discussion
5,6,7 Note included. The abstract includes the conclusion at the end of the article.
Reviewer 3 Report
In general, this is not really a scientific study, but rather a review of vaccination policies. Since the focus of the special issue is the economic aspects of vaccination, the manuscript is appropriate for that reason. The presentation is clear, there are no language issues. Specific comments:
The Introduction does not state the purpose of the study, or hypothesis, if there was one.
The Methods section is short, but there is probably not much more to say there.
Results: clear and easy to follow. Adding data from another area, for example the US, or EU in general would be interesting for the reader for comparison. This would help to understand the meaning of the data presented here. Some of this is done in the discussion, however, very limited, and it would be easier to see them head to head and more detailed.
Discussion: See above. The discussion is mostly general statements, not directly related to the results. Nonetheless, some of the findings are discussed appropriately, but not all of them.
The conclusions are otherwise supported with the results.
In summary, the manuscript seems to be appropriate for this special issue, pending some improvements.
Author Response
1. The introduction states the purpose of the study:
Objective
The objective of the study is to analyse health policy programs including immunization programs, which were developed, implemented and financed by local government units of all levels in Poland between 2016 and 2019.
2. Unfortunately, we cannot take this comment into account because the data presented in the article represent only information regarding the financing of vaccinations by local government units. They do not include information on the number of all vaccinated persons in Poland.
3.
Reviewer 4 Report
Actually, this does not appear as a scientific paper. To me, it sounds closest to an administrative report by the Ministry of Health to the Government: that much allocated, that much spent, that many vaccinations performed. For instance, vaccinated people are always absolute numbers, almost never a proportion of the target population. There are no informations about the cost per vaccine injected in the different situations, to show how the money has been spent. The only, very general, information that the reader obtains is that different amounts of money have been spent at different levels; that the results obtained in the different realities are very different; that the people reached by the different vaccines are growing in number (and probably as proportions of the population, but nothing about the amount of the proportion!), that there are dramatic differences in results between the different levels of responsibilities (poviats, communes), but nobody can understand which are the trends of vaccinations, which is the vaccination obtaining better results. It is clear, finally, that papilloma vaccination in young girls and influenza vaccination at the target ages is very far from optimal.
Author Response
- Because vaccination programs are implemented at different levels of local government organization (provinces, districts, municipalities) it is not possible to calculate unequivocally the percentage of vaccinated persons in the population at each of the indicated levels of local government. It is not possible to distinguish groups that would be a point of reference for such calculations. The inhabitants of a municipality may participate in programs implemented by the municipality in which they live as well as in programs implemented by the county or province where the municipality is located.
- The information contained in the reports of the provincial governors does not allow us to determine what portion of the funds was spent on financing immunizations and what portion was spent on other activities such as promotion.
Round 2
Reviewer 3 Report
The authors provided a response organized in three bullets. The third one is empty. I don't see any changes the authors made in response to the reviewer's comments. Hence, there's nothing to review.
Author Response
We prepare objective
Objective
The objective of the study is to analyse health policy programs including immunization programs, which were developed, implemented and financed by local government units of all levels in Poland between 2016 and 2019.
Results: Unfortunately, we cannot take this comment into account because the data presented in the article represent only information about the financing of vaccinations by local government units. They do not include information on the number of all vaccinated persons in Poland.
Discussion: We prepare a new discussion, we added:
There has been a disproportionately high commitment of financial resources in relation to the number of vaccinated people. This probably results from the implementation of promotional campaigns within the programmes using the most expensive media, i.e. radio, TV. Conducting promotional campaigns within the programmes is absolutely justified, however, the effectiveness of promotional campaigns involving very high financial resources seems highly debatable, taking into account the small number of vaccinated people. It is also worth noting that the information contained in the reports prepared by voivodes does not allow for the determination of which part of the financial resources was allocated to the financing of preventive vaccinations, and which to promotional campaigns.
The conclusions are supported by the results .